# Assessment of the Carbon Emission Reduction Effect of the Air Pollution Prevention and Control Action Plan in China

**DOI:** 10.3390/ijerph182413307

**Published:** 2021-12-17

**Authors:** Zhenhua Zhang, Jingxue Zhang, Yanchao Feng

**Affiliations:** 1Institute of Green Finance, Lanzhou University, Lanzhou 730000, China; zhangzhenhua@lzu.edu.cn; 2Business School, Zhengzhou University, Zhengzhou 450001, China; 202011011010004@gs.zzu.edu.cn

**Keywords:** carbon emission reduction, Air Pollution Prevention and Control Action Plan, parallel trend test, placebo test, spatial difference-in-differences model

## Abstract

In this study, we propose an integrated econometric framework incorporating the difference-in-differences model, the propensity-score-matching difference-in-differences model, and the spatial difference-in-differences model to explore the effect of the Air Pollution Prevention and Control Action Plan on per capita carbon emission in China at the national, regional, and administrative levels. Contradictory results are supported under different econometric models, which highlight the importance and necessity of comprehensive analysis. Taking 285 prefecture-level and above cities as an example, the empirical results show that APPCAP has effectively reduced per capita carbon emission in China at the national level without the consideration of the spatial spillover effect. However, with the consideration of the spatial spillover effect, APPCAP has effectively and directly increased per capita carbon emission in local pilot cities at the national level, and reduced it among pilot cities via the spatial spillover effect, but the effects have become invalid in the non-pilot cities neighboring the pilot cities. Furthermore, the spatial heterogeneity of the effects of APPCAP on per capita carbon emission are supported at the regional and administrative levels. Finally, some specific policy implications are provided for achieving the “win-win” situation of energy saving, emission reduction, and economic development.

## 1. Introduction

In the 21st century, accompanied by the excessive consumption of fossil energies and the ecological deterioration of air quality, the conflict between economic development and environmental protection has become a bottleneck for China’s sustainable development, that is, the extensive and high-speed economic development mode has lost its momentum and should be converted to high-quality development in the new era [1]. In addition, the severe shock of COVID-19 has worsened the international economic and investment environment, that is, the dual pressures—domestic and from overseas—also called for the transformation of development from extensive to intensive [2]. Thus, balancing the relationship between ecological civilization and steady economic development has become a great challenge for the Chinese government [3].

Specifically, to abate the severe air pollution and promote green development, the State Council of China released the Air Pollution Prevention and Control Action Plan (i.e., APPCAP) on 10 September 2013 [4]. Considered as the first comprehensive plan to control air pollution in China, the APPCAP contains a series of stringent measures, including discharge standards, monitoring plans, and accountability systems [5]. Since its implementation, the APPCAP in China has gained enough attention in academia, and a basic consensus has been reached on its effectiveness in improving air quality, which provides a valuable insight into the pollution abatement effect of it, while few studies have paid attention to how this policy affects carbon emission [6].

The rapid growth of carbon emissions has led to global climate change, which forms an enormous challenge to human sustainable development [7]. Against this background, in order to shoulder the international responsibility and promote the construction of “A Community of Shared Future for Mankind”, Chinese president Xi Jinping put forward the program of “reaching carbon dioxide emissions peak before 2030 and achieving carbon neutrality before 2060” at the 75th United Nations General Assembly on 22 September 2020 [8]. Then, China constructed a national carbon online trading market on 16 July 2021, which will play a vital role in China’s energy saving and emission reduction strategy. Thus, learning from the past by exploring the effect of APPCAP on carbon emission is of great significance to academia and practice. However, the spatial spillover effect and spatial heterogeneity are often ignored in the empirical analysis of policy evaluation, which delivers another research gap for this study [9].

Therefore, the main contributions of this study lie in three aspects. Firstly, an integrated econometric framework incorporating the difference-in-differences (i.e., DID) model [10], the propensity-score-matching difference-in-differences (i.e., PSM-DID) model [11], and the spatial difference-in-differences (i.e., SDID) model [12] is proposed to explore the effect of APPCAP on per capita carbon emission in China, which can provide more specific policy recommendations for reducing carbon emissions. Secondly, to gain a deep insight into the spatial heterogeneity, this study divided the full sample into subgroups at the regional and administrative levels, which is conducive to improving the significance in practice [13]. Finally, as a typical emerging country, the effect of APPCAP on per capita carbon emission in China can provide an instructive policy reference for other developing counties in similar situations in terms of transforming their economic development mode from extensive to intensive.

The rest of this study is constructed as follows. Section 2 provides the policy background of the APPCAP. Section 3 introduces the methodology, including variables’ selection, data sources, and econometric models. Section 4 reports the estimation results of the DID, PSM-DID, and SDID models, as well as the parallel trend test and the placebo test. Section 5 discusses the spatial heterogeneity at the regional and administrative level. Section 6 concludes this study and provides policy implications.

## 2. Policy Background of the APPCAP

To alleviate the health risks and economic losses caused by severe air pollution, the Chinese central government has implemented a series of air pollution prevention and control policies, such as the 11th Five-Year Plan on Environmental Protection (2006–2010), the Energy Conservation and Emissions Reduction policy during the 11th Five-Year Plan, the 12th Five-Year Plan on Environmental Protection (2011–2015), the 12th Five-Year Plan on Energy Conservation and Emissions Reduction, the 12th Five-Year Plan on Air Pollution Prevention and Control in Key Regions, and the 12th Five-Year Plan on Air Pollution Prevention and Control [14].

In particular, the goals of air pollutant concentration control were initially proposed in the 12th Five-Year Plan on Air Pollution Prevention and Control in Key Regions issued on 29 October 2012, and strengthened in the 12th Five-Year Plan on Air Pollution Prevention and Control Action Plan issued on 10 September 2013 [15,16]. As the first comprehensive plan for air pollution prevention and control, APPCAP comprises a series of detailed contents, including ten key actions and 35 concrete measures, detailed assessment plans and clear accountability models for local governments, and substantial measures and supporting policies [5].

As a milestone for air quality control in China, APPCAP achieved great success in PM2.5 (fine particulate matter with a diameter less than 2.5 μm) and PM_10_ (inhalable particles with a diameter less than 10 μm) concentration reductions during the first five years [4,17]. For instance, five years after the peak in 2013, the annual average concentrations of PM_2.5_ decreased by 39%, 34%, and 26% in the Beijing–Tianjin–Hebei Region, the Yangtze River Delta Region, and the Pearl River Delta Region, respectively [5]. Correspondingly, the annual average concentrations of PM_10_ in those three regions decreased by 19%, 28%, and 22%, respectively [5].

Since carbon emission plays a vital role in China’s battle against air pollution, the air pollution policies during 11th Five-Year Plan mainly focused on total emission reduction, and this trend was basically extended during 12th Five-Year Plan [14]. The implementation of APPCAP indicates that the goals of air pollution control began to focus on the environmental quality, while the success of this policy for air pollution control has been confirmed by many studies [15,16,17,18]. Therefore, considering the inertia of the emission control-oriented policy, the questions as to whether and how APPCAP affects carbon emission still deserve in-depth research against the background of carbon peak and carbon neutralization.

## 3. Methodology

### 3.1. Variables Selection

#### 3.1.1. Dependent Variable

Referring to the study of Chen et al. (2020) [19], this study first calculated provincial CO_2_ emissions based on the provincial energy balance tables and established the relationship between provincial CO_2_ emissions and nighttime light data (i.e., the sum of the Digital Number (i.e., DN) values from the Defense Meteorological Satellite Program/Operational Linescan System (i.e., DMSP/OLS) data and the National Polar-orbiting Partnership/Visible Infrared Imaging Radiometer Suite (i.e., NPP/VIIRS) data; then, the sum of the DN values was employed as a proxy to estimate the prefecture-level carbon emissions. Finally, to alleviate the influence of population flow and growth, this study adopted per capita carbon emission as the proxy of dependent variable.

In particular, based on the methods in the 2006 Guidelines for National Greenhouse Gas Inventories provided by the Intergovernmental Panel on Climate Change (i.e., IPCC), provincial CO_2_ emissions could be calculated by the following formula [19]:(1)$$CO_{2}=\sum_{i=1}^{4}E_{it}\times NCV_{i}\times CEF_{i}\times COF_{i}\times (44/12)$$
where *i* = 1, 2, 3, 4 represent coal, oil, natural gas, and non-fossil fuels, respectively. *E_it_* denotes the *i_th_* type of energy consumption, *NCV_i_* denotes the average low calorific value of the *i_th_* energy source, *CEF_i_* denotes the carbon content of the *i_th_* energy source, *COF_i_* denotes the carbon oxidation factor of the *i_th_* energy source, and 44 and 12 denote the relative atomic mass of carbon dioxide and carbon, respectively.

#### 3.1.2. Key Explanatory Variables

Two dummy indicators, such as post and treat, were employed to act as the proxy of key explanatory variables, and their interaction term (i.e., *treat × post*) is employed to act as the proxy of APPCAP. Post is the time dummy variable, which equals 1 after 2013, and 0 otherwise. Treat is the city dummy variable, which equals 1 when the city *i* is in the treatment group, and 0 otherwise.

#### 3.1.3. Control Variables

In addition to key explanatory variables, several control variables were introduced into this study to capture the influences of other factors on carbon emission in China. The control variables included fiscal decentralization (*FD*, defined by the proportion of financial expenditure to financial revenue), industrial upgrading (*IU*, defined by the proportion of the added value of the tertiary industry to the secondary industry), urbanization rate (*UR*, defined by the proportion of non-farm population to city’s total population), and foreign direct investment (*FDI*, defined by the proportion of foreign direct investment to GDP), which were employed by referring to the studies of [9,10,13], etc.

### 3.2. Data Sources

In 2018, China promulgated an updated version of APPCAP, that is, the Three Year Action Plan to Beat Air Pollution, which aims to further strengthen the joint prevention and control of air pollution [16]. Thus, to eliminate the effect of the Three Year Action Plan to Beat Air Pollution, the sample of this study included 285 prefecture-level and above cities in China from 2007 to 2017, excluding the cities dismantled and established during the research period, and the cities with severe data loss and inaccessibility. In addition, the interpolation method was employed to supplement the missing values of individual cities. In particular, 47 prefecture-level and above cities in the Beijing–Tianjin–Hebei Region, the Yangtze River Delta Region, the Pearl River Delta Region, and the Fen-Wei Plain were set as the treatment group, and another 238 prefecture-level and above cities were set as the control group [20]. As for the data sources, several channels were adopted in this study. For instance, the provincial data of energy use were derived from the China Energy Statistics Yearbook; the nighttime light brightness data of DMSP/OLS and NPP/VIIRS were derived from the corresponding websites of http://ngdc.noaa.gov/eog/dmsp/downloadV4composites.html and https://www.ngdc.noaa.gov/eog/viirs/download_dnb_composites.html, respectively (accessed on 21 November 2020); the list of APPCAP were collected from the 12th Five-Year Plan on Air Pollution Prevention and Control in Key Regions; while the socioeconomic data of control variables were derived from the China City Statistical Yearbook. The statistical description of the whole sample is reported in Table 1.

### 3.3. Econometric Models

One important observation is that APPCAP caused a difference between the treatment group and the control group. Another is that APPCAP created a difference between the treatment group before and after its implementation. Thus, APPCAP could be treated as a quasi-natural experiment, and the difference-in-differences (i.e., DID) model could effectively identify the effect of it on per capita carbon emission in China [10]. Therefore, the benchmark DID model was defined as:(2)$$Y_{it}=\beta_{0}+\beta_{1}\times treat_{i}\times post_{t}+\beta_{2}\times control_{it}+\mu_{i}+\lambda_{t}+\varepsilon_{it}$$
where *Y_it_* denotes the per capita carbon emission of city *i* in year *t*, *treat_i_* denotes the city dummy variable, *post_t_* denotes the time dummy variable, *control_it_* denote a vector of control variables, *β*_0_ denotes the constant term, *β*_1_ denotes the policy coefficient, *β*_2_ denotes the coefficients of control variables, *μ_i_* denotes the city fixed effect, *λ_t_* denotes time fixed effect, *ε_it_* denotes the random interference term.

To reduce the differences between the treatment group and the control group, this study also employed the propensity-score-matching difference-in-differences (i.e., PSM-DID) method to test the effect of APPCAP on carbon emission in China [11]. In particular, the propensity-score-matching (i.e., PSM) radius matching method was employed to estimate the propensity scores, the dependent variable was per capita carbon emission, and the control variables formed the covariate of PSM. After PSM matching, the anomalous samples were deleted, that is, there was no significant systematic difference between the treatment group and the control group, and thus, the differences in per capita carbon emission could only be caused by APPCAP.

To simultaneously investigate the direct and indirect effects of APPCAP on per capita carbon emission in China, this study also employed the spatial difference-in-differences (i.e., SDID) model by referring to the research of Chagas et al. (2016) [12]. Specifically, the SDID model is constructed as follows:(3)$$\begin{array}{l} Y_{it}=\beta_{0}+\beta_{1}\times treat_{i}\times post_{t}+\beta_{2}\times control_{it}+\gamma_{1}\times W_{T,T}D_{it}+\gamma_{2}\times W_{NT,T}D_{it}\\ +\gamma_{3}\times W\times control_{it}+\mu_{i}+\lambda_{t}+\varepsilon_{it} \end{array}$$
where *W_T,T_D_it_* denotes the spatial spillover effects on the pilot cities, *W_NT,T_D_it_* denotes the spatial spillover effects on the non-pilot samples neighboring the pilot cities, *W* denotes the squared inverse distance weight matrix [9]. *γ*_1_, *γ_2_*, and *γ_3_* denote the spatial coefficients of *W_T,T_D_it_*, *W_NT,T_D_it_*, and *W × control_it_*, respectively. Other parameters are defined as described in Equation (2).

## 4. Empirical Results and Analysis

### 4.1. Parallel Trend Test

Before regression, the key identification assumption of the DID model was that the carbon emission in non-pilot cities provide effective counterfactual changes to the carbon emission in pilot cities. However, a potential challenge of the above-mentioned assumption is the variability between the pilot and non-pilot cities driven by pre-existing time trends. Thus, to alleviate this concern, two diagnostic tests were employed to verify that this assumption was not violated, that is, before the implementation of the APPCAP, the per capita carbon emission of each pilot city should have maintained a relatively steady trend of change.

In the first diagnostic test, following the method of Hu et al. (2020) [21], the yearly changes in the mean values of per capita carbon emission between pilot cities and non-pilot cities were observed. As shown in Figure 1, before 2013, the pilot cities and non-pilot cities had similar time trends in terms of per capita carbon emission, while this trend changed after 2013 following the implementation of the APPCAP pilot policy.

In the second diagnostic test, following the method of Feng et al. (2021) [10], we constructed a series of time dummy variables to capture the dynamic effect of APPCAP on per capita carbon emission. If the pilot cities and non-pilot cities had similar time trends before 2013, the coefficients of *treat* × *year* were expected to be statistically insignificant. In particular, we constructed the following regression:(4)$$Y_{it}=\beta_{0}+\sum_{t=2007}^{2017}\beta_{t}\times treat_{i}\times year_{t}+\beta_{2}\times control_{it}+\mu_{i}+\lambda_{t}+\varepsilon_{it}$$
where *year_t_* denotes a series of time dummy variables, and other parameters are also defined as described in Equation (2). As shown in Figure 2, the coefficients of *treat* × *year* were statistically insignificant before 2013, and became significantly negative from 2013 onwards; thus, there was no systematic difference found between pilot cities and non-pilot cities in the absence of the APPCAP pilot policy. Therefore, the parallel trend assumption was satisfied, and the DID approach was found to be suitable for this study.

### 4.2. Placebo Test

To alleviate the concern of other unknown factors on the selection of pilot cities, we performed the placebo test by randomly selecting several virtual treatment groups to estimate the benchmark DID model, and to ensure that the conclusion of this study was caused by the implementation of APPCAP [18]. In particular, random sampling was performed 1000 times among full samples, with 47 prefecture-level and above cities randomly selected as the treatment group, while the other 238 prefecture-level and above cities were selected as the control group; the kernel density distribution is illustrated in Figure 3. In particular, the *x*-axis represents the policy coefficient’s *t*-value of *treat* × *year*, and the *y*-axis represents the corresponding *p*-value. As can be seen from Figure 3, almost all the absolute *t*-values of the randomly sampling estimation coefficients were less than 2 and their corresponding *p*-values were greater than 0.1, which indicates that the unobserved factors had a negligible impact on the estimation results; thus, the placebo test was also not violated.

### 4.3. Estimation Results of the DID Model

Based on Equation (2), this study adopted the DID model with a time fixed effect to explore the effect of APPCAP on per capita carbon emission in China; the parameter estimation results are reported in column (2) of Table 2, and the estimation results in column (1) are selected as a comparison without any control variables. It can be seen that the coefficients of the key explanatory variable *treat × post* are significantly negative in columns (1) and (2), indicating that the implementation of APPCAP reduced the per capita carbon emission in China, and this was found to be the case with or without the consideration of control variables. The reason for this may be that APPCAP was able to promote the green transformation by optimizing the energy structure, thus limiting high-pollution industrial activity and reducing per capita carbon emissions simultaneously.

For the control variables, UR and FDI were found to be positively correlated to per capita carbon emission at the significance levels of 1% and 5%, respectively. This finding shows that the process of urbanization and the inflow of FDI can inevitably increase per capita carbon emission, which, to some extent, verifies the “Pollution Heaven Hypothesis” of China, that is, enterprises in pollution-intensive industries tend to transfer into the countries or regions with relatively low environmental standards. FD and IU were positively correlated to per capita carbon emission but at an insignificant level. This shows that fiscal decentralization and industrial upgrading played an invalid role in terms of affecting per capita carbon emission, which highlights the complexity and difficulty of carbon emission reduction. On one hand, the improvement in production efficiency caused by fiscal decentralization and industrial upgrading may lead to a reduction in per capita carbon emission. On the other hand, the rise in production scale caused by fiscal decentralization and industrial upgrading may lead to the energy rebound effect and ultimately cause an increase in per capita carbon emission. Thus, the insignificant coefficients of FD and IU indicate that those two powers are even-handed.

### 4.4. Estimation Results of the PSM-DID Model

To ensure the robustness of DID analysis results, this study employed the PSM-DID model to re-investigate the effect of APPCAP on per capita carbon emission in China. The validity test results for tendency score matching are reported in Table 3. It can be found that all the variables were not significant at the level of 10% after PSM, that is, the null hypothesis of no significant difference between the treatment group and the control group was accepted, and PSM-DID was effective. In addition, as shown in Figure 4, the covariate standardization bias test indicated that only a small amount of the samples failed to match, which also to some extent supported the effectiveness the PSM-DID model.

The parameter estimation results of the PSM-DID model are reported in column (2) of Table 4, and the estimation results in column (1) are selected as a comparison without any control variables. It is indicated that all the coefficients did not change significantly after PSM, and this was found to be the case with or without the consideration of control variables. Therefore, APPCAP indeed reduced per capita carbon emission, that is, the results of this study have high-level robustness.

### 4.5. Estimation Results of the SDID Model

Based on the Equation (3), this study explored the direct and indirect effects of APPCAP on per capita carbon emission in China by adopting the SDID model under the squared inverse distance weight matrix; the corresponding results are reported in column (2) of Table 5, and the estimation results in column (1) are selected as a comparison without any control variables. It can be found that the direct coefficients of *treat × post* are significantly positive, the spatial coefficients of *W_T,T_D* are significantly negative, and the spatial coefficients of *W_NT,T_D* are positive but insignificant in columns (1) and (2), indicating that the implementation of APPCAP increased per capita carbon emission in local pilot cities, but reduced per capita carbon emission among pilot cities via the spatial spillover effect, while the spatial spillover effect on reducing per capita carbon emission was not supported in the non-pilot cities neighboring the pilot cities. Thus, in contrast to the non-spatial DID and PSM-DID models, the SDID model effectively identified the channel of APPCAP in reducing per capita carbon emission from the spatial spillover effect among the pilot cities, which highlights the advancement and innovation of this study.

## 5. Heterogeneous Analysis

### 5.1. Regional Heterogeneity Analysis

Referring to the geographical division standard announced by the Chinese National Bureau of Statistics (CNBS), this study divided the full sample into three regions, namely the eastern region, the central region, and the western region. Then, considering the advantage of the SDID model in identifying the spatial spillover effect, this study employed it to conduct the estimation at the regional level; the corresponding results are reported in columns (2), (4), and (6) of Table 6, and the estimation results in columns (1), (3), and (5) are selected as a comparison without any control variables. It can be found that the direct coefficients of *treat*
*×*
*post* are significantly positive in columns (1)–(6), the spatial coefficients of *W_T,T_D* are significantly negative in columns (1), (2), (5), and (6), but insignificant in columns (3) and (4), and the spatial coefficients of *W_NT,T_D* are significantly positive in columns (5) and (6), but insignificant in columns (1)–(4), indicating that the implementation of APPCAP also increased per capita carbon emission in local pilot cities at the regional level, but merely reduced per capita carbon emission among pilot cities via the spatial spillover effect in the eastern and western regions, and increased per capita carbon emission in the non-pilot cities neighboring the pilot cities of the western region. After all, most of the pilot cities are located in the eastern region; thus, the estimation results, to some extent, verified the rationalization of the regional analysis. Meanwhile, the spatial heterogeneity of APPCAP affecting per capita carbon emission is supported at the regional level.

### 5.2. Administrative Heterogeneity Analysis

To investigate the heterogeneous effect of APPCAP on per capita carbon emission at the administrative level, this study divided the full samples into two parts: first- and second-tier cities and third-tier cities. Specifically, the first- and second-tier cities are the centrally administrated municipalities, provincial capitals, and sub-provincial cities, while the third-tier cities are the prefecture-level cities. Then, considering the advantage of the SDID model in identifying the spatial spillover effect, this study employed it to conduct the estimation at the administrative level; the corresponding results are reported in columns (2) and (4) of Table 7, and the estimation results in columns (1) and (3) are selected as a comparison without any control variables. It can be found that the direct coefficients of *treat × post* are significantly positive in columns (1) and (2), and positive but insignificant in columns (3) and (4), the spatial coefficients of *W_T,T_D* are significantly negative in columns (1) and (2), significantly positive in column (3), and positive but insignificant in column (4), and the spatial coefficients of the spatial coefficients of *W_NT,T_D* are significantly positive in column (3), negative but insignificant in column (1), and positive but insignificant in columns (2) and (4), indicating that the implementation of APPCAP not only increased per capita carbon emission in local pilot cities of the first- and second-tier cities, but also reduced per capita carbon emission among pilot cities via the spatial spillover effect of the first- and second-tier cities, while the direct and indirect effects of APPCAP on per capita carbon emission of the third-tier cities were relatively invalid and unstable. Thus, the spatial heterogeneity of APPCAP affecting per capita carbon emission is also supported at the administrative level, and the popularization of extending it in the third-tier cities still has a great space to improve.

## 6. Conclusions and Policy Implications

### 6.1. Conclusions

In this study, we proposed a comprehensive research framework including the DID, PSM-DID, and SDID models to explore the effects of APPCAP on per capita carbon emission in China at the national, regional, and administrative levels. Taking 285 prefecture-level and above cities from 2007 to 2017 as an example, this study empirically examined the aforementioned effects and draws the following conclusions.

Firstly, without the consideration of the spatial spillover effect, it was found that APPCAP effectively reduced per capita carbon emission in China at the national level, while two control variables including *UR* and *FDI* had an opposite effect on it, which, to some extent, verified the “Pollution Heaven Hypothesis” of China.

Secondly, with the consideration of the spatial spillover effect, it was found that APPCAP effectively and directly increased per capita carbon emission in local pilot cities at the national level, and reduced it among pilot cities via the spatial spillover effect, but became invalid in the non-pilot cities neighboring the pilot cities.

Thirdly, the effects of the spatial heterogeneity of APPCAP on per capita carbon emission were supported at the regional and administrative levels, and the effects of APPCAP in terms of reducing per capita carbon emission in pilot cities were established in the eastern cities and the first- and second-tier cities via the spatial spillover effect.

### 6.2. Policy Implications

According to the above-mentioned findings, several policy implications are proposed as follows.

Firstly, to achieve the win-win of emission reduction and economic development in China, the traditional development path of “pollution first and control later” should turn to the sustainable development path of “prioritizing ecological conservation and boosting green development”, such as eliminating backward production capacity, promoting the use of renewable energy in the process of urbanization, and improving the threshold and quality of *FDI*.

Secondly, to enjoy the bonus of the spatial spillover effect in China, the synergy between APPCAP and a national carbon trading system should be supported to reduce the cost of carbon trading and improving carbon trading efficiency. In addition, the cooperation between the carbon market and environmental policies could enhance the transparency of various entities to guarantee the fairness of cleaner production and the efficiency of green innovation.

Thirdly, regional and administrative heterogeneity should be considered when formulating environmental policies, and goals specific to different stages rather than rigid policy provisions should be formulated to satisfy local conditions and characteristics. In addition, green transformation should be encouraged for high-pollution and high-emission enterprises rather than transferred to less developed cities, and more financial funds should be invested to promote green development.

Although it attempted to investigate the effect of APPCAP on per capita carbon emission in China by utilizing several econometric models, this study still has some limitations, which deserve in-depth research in the future. For instance, the dynamic effect was ignored in the empirical analysis, which, to some extent, limits the practical significance of this study. Even so, this study attained its original goal and obtained diverse interesting findings.

## Figures and Tables

**Figure 1 ijerph-18-13307-f001:**
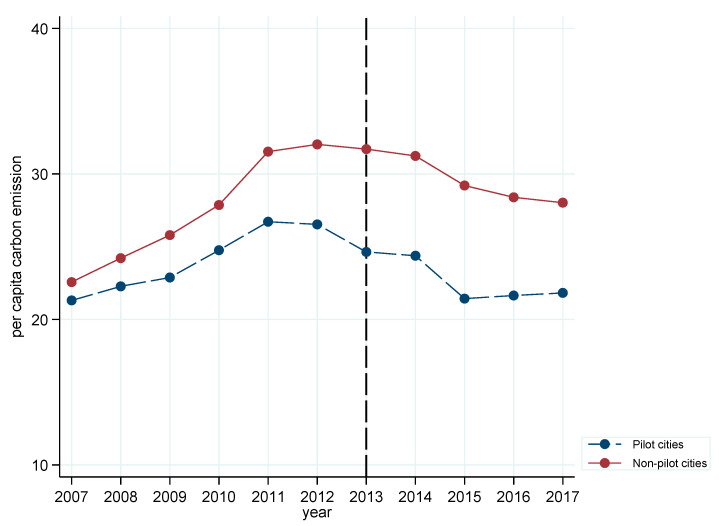
Annual average of per capita carbon emission.

**Figure 2 ijerph-18-13307-f002:**
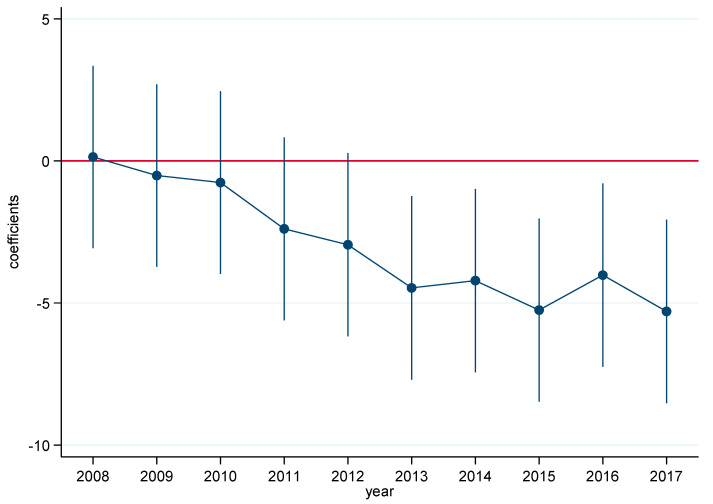
Box plot of parallel trend test.

**Figure 3 ijerph-18-13307-f003:**
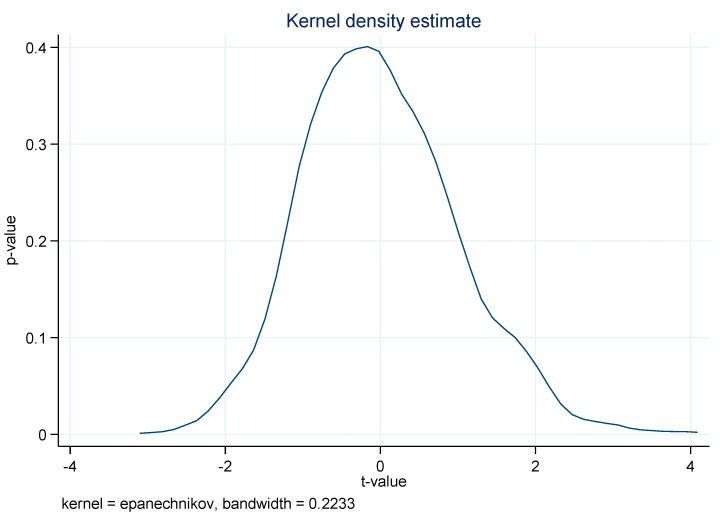
The graph of the policy estimation coefficient of 1000 random sampling experiment panels with a bandwidth of 0.2233 (fixed model).

**Figure 4 ijerph-18-13307-f004:**
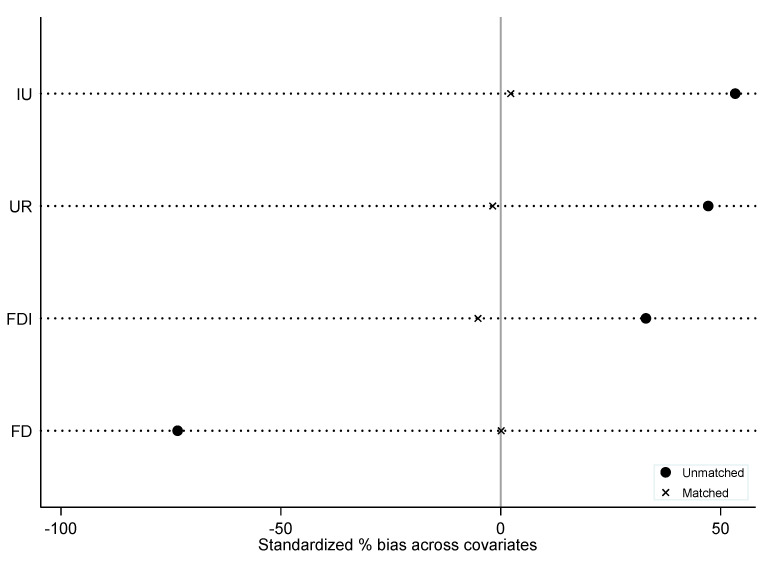
Covariate standardization bias test.

**Table 1 ijerph-18-13307-t001:** Statistical description.

Variables	Observations	Mean	S.D.	Min	Max
CO_2_	3135	27.606	30.193	1.875	405.582
Post	3135	0.545	0.498	0.000	1.000
Treat	3135	0.165	0.371	0.000	1.000
FD	3135	2.353	2.063	0.000	43.844
IU	3135	0.869	0.481	0.094	5.340
UR	3135	0.597	0.255	0.046	1.000
FDI	3135	0.021	0.030	0.000	0.775

**Table 2 ijerph-18-13307-t002:** Parameter estimation results of the DID model.

Variables	DID	
	(1)	(2)
*post × treat*	−3.886 ***	−3.649 ***
	(−5.444)	(−5.175)
*FD*		0.083
		(0.810)
*IU*		0.138
		(0.215)
*UR*		11.474 ***
		(11.026)
*FDI*		14.363 **
		(2.457)
Constant	22.360 ***	17.738 ***
	(51.109)	(21.722)
Observations	3135	3135
R-squared	0.142	0.178

Note: *t* statistics in parentheses; *** *p* < 0.01, ** *p* < 0.05.

**Table 3 ijerph-18-13307-t003:** Validity test results for tendency score matching.

Variables	Unmatched	Mean		%Reduct		*t*-Test		V(T)/V(C)
	Matched	Treated	Control	%bias	|bias|	*t*	*p* > |t|	
*FD*	U	1.32	2.46	−73.50		−8.92	0.00	0.05 *
	M	1.32	1.32	0.10	99.90	0.03	0.98	0.94
*IU*	U	1.12	0.84	53.30		9.31	0.00	1.48 *
	M	1.07	1.06	2.20	95.80	0.24	0.81	0.37 *
*UR*	U	0.70	0.59	47.20		7.17	0.00	0.76 *
	M	0.70	0.70	−1.90	96.10	−0.23	0.82	0.86
*FDI*	U	0.03	0.02	33.00		6.32	0.00	2.15 *
	M	0.03	0.03	−5.20	84.30	−0.88	0.38	0.60 *

Note: z-statistics in parentheses; * *p* < 0.1.

**Table 4 ijerph-18-13307-t004:** Parameter estimation results of the PSM-DID model.

Variables	PSM-DID	
	(1)	(2)
*post × treat*	−3.794 ***	−3.467 ***
	(−6.430)	(−5.940)
*FD*		0.219
		(0.855)
*IU*		0.105
		(0.136)
*UR*		11.576 ***
		(8.443)
*FDI*		18.234 **
		(2.083)
Constant	28.419 ***	20.481 ***
	(200.755)	(16.213)
Observations	3015	3015
R-squared	0.946	0.948

Note: *t* statistics in parentheses; *** *p* < 0.01, ** *p* < 0.05.

**Table 5 ijerph-18-13307-t005:** Parameter estimation results of the SDID model.

Variables	SDID	
	(1)	(2)
*post × treat*	6.404 ***	6.367 ***
	(12.451)	(12.437)
*W_T,T_D*	−5.753 ***	−6.083 ***
	(−8.371)	(−8.822)
*W_NT,T_D*	0.332	0.018
	(0.588)	(0.033)
Control variables	No	Yes
Observations	3135	3135
R-squared	0.541	0.548

Note: *t* statistics in parentheses; *** *p* < 0.01.

**Table 6 ijerph-18-13307-t006:** Regional heterogeneity results of the SDID model.

Variables	Eastern		Central		Western	
	(1)	(2)	(3)	(4)	(5)	(6)
*post × treat*	4.420 ***	3.992 ***	6.155 **	5.295 *	10.308 ***	10.076 ***
	(5.141)	(4.605)	(2.024)	(1.744)	(13.431)	(13.068)
*W_T,T_D*	−3.743 ***	−3.410 ***	602.672	2014.109	−5.927 ***	−6.414 ***
	(−3.734)	(−3.276)	(0.142)	(0.474)	(−4.146)	(−4.375)
*W_NT,T_D*	−0.780	−1.008	0.021	−0.184	3.435 ***	2.800 ***
	(−0.785)	(−1.010)	(0.017)	(−0.149)	(4.057)	(3.272)
Control variables	No	Yes	No	Yes	No	Yes
Observations	1111	1111	1199	1199	825	825
R-squared	0.518	0.530	0.563	0.571	0.624	0.634

Note: *t* statistics in parentheses; * *p* < 0.1, ** *p* < 0.05, *** *p* < 0.01.

**Table 7 ijerph-18-13307-t007:** Administrative heterogeneity results based on the SDID model.

Variables	First- and Second-Tier Cities		Third-Tier Cities	
	(1)	(2)	(3)	(4)
*post × treat*	2.913 **	3.930 ***	1.031	0.957
	(2.116)	(2.893)	(1.169)	(1.097)
*W_T,T_D*	−11.008 ***	−9.483 ***	1.784 *	1.235
	(−6.976)	(−5.815)	(1.709)	(1.192)
*W_NT,T_D*	−36.629	1.194	0.870 *	0.519
	(−1.528)	(0.048)	(1.842)	(1.105)
Control variables	No	Yes	No	Yes
Observations	385	385	2750	2750
R-squared	0.631	0.666	0.565	0.578

Note: *t* statistics in parentheses; * *p* < 0.1, ** *p* < 0.05, *** *p* < 0.01.

## Data Availability

The data used to support the findings of this study are available from the corresponding author upon request.

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
