# Peer review of "Assessment of the Carbon Emission Reduction Effect of the Air Pollution Prevention and Control Action Plan in China"

_ijerph, 2021, doi:10.3390/ijerph182413307_

Round 1

Reviewer 1 Report

This manuscript seeks to examine the effectiveness of APPCAP on the carbon emission at the prefecture- and above city-level in China. This is a critical topic to empirically evaluate the effectiveness of a governmental intervention to reduce carbon emission, which is the primary goal worldwide.

Luckily enough, China currently has built a natural experiment by implementing APPCAP in part of prefectures and above cities, and DID, PSM DID, and Spatial DID should be sound experimental design to examine the causality/effectiveness of the Plan on carbon emission. However, current models miss post- or treatment-variable controlling the main effect of each variable from the model and isolate the interaction term of the two, which is the actual experiment—DID--effect. Also, the results from the three models are inconsistent. Author(s) did not sufficiently explain them; instead, it is asserted that spatial spillover effects sort of offset the Plan’s effect. I would suggest including both post- and treatment-variable in the model and see if results are changed.

This manuscript measures the carbon emission based on energy consumption but does not address it enough in the body of literature. Please give more details on the measurement of DV.

Author Response

Responses to Reviewer # 1

Comments and Suggestions for Authors

This manuscript seeks to examine the effectiveness of APPCAP on the carbon emission at the prefecture- and above city-level in China. This is a critical topic to empirically evaluate the effectiveness of a governmental intervention to reduce carbon emission, which is the primary goal worldwide.

Luckily enough, China currently has built a natural experiment by implementing APPCAP in part of prefectures and above cities, and DID, PSM DID, and Spatial DID should be sound experimental design to examine the causality/effectiveness of the Plan on carbon emission. However, current models miss post- or treatment-variable controlling the main effect of each variable from the model and isolate the interaction term of the two, which is the actual experiment—DID--effect. Also, the results from the three models are inconsistent. Author(s) did not sufficiently explain them; instead, it is asserted that spatial spillover effects sort of offset the Plan’s effect. I would suggest including both post- and treatment-variable in the model and see if results are changed.

This manuscript measures the carbon emission based on energy consumption but does not address it enough in the body of literature. Please give more details on the measurement of DV.

Reply: Many thanks for your constructive advice. (1) Contradictory results are supported under different econometric models, which highlights the importance and necessary of comprehensive analysis. (2) Followed your guidance, we have added both post- and treatment-variable in the model and re-conducted the estimation. The changes of results are negligible, while some coefficients of the post- and treatment-variable are unattainable, one possible reason is that there too many zeros in those variables. Thus, we attempts to improve the prior models by employing the dual fixed equation, and replaced the former results with the new results. After comparison, we found that the changes are negligible, which to some extent proved that the results of this paper are robust and reliable. (3) As for the measurement of dependent variable, we have added more details in the text.

Above are the detailed corrections we made according to your comments point by point. Thanks again for your meticulous review and valuable suggestions to improve our manuscript. We hope that the revised revision has addressed all the issues. We are looking forward to your positive response. If you have any queries, please don’t hesitate to contact me at the address below.

Best regards,

Zhenhua Zhang1, Jingxue Zhang2 and Yanchao Feng2*

1 Institute of Green Finance, Lanzhou University, Lanzhou 73000, PR China

2 Business School, Zhengzhou University, Zhengzhou 450001, PR China

* Correspondence: m15002182995@163.com; Tel.: +86-150-0218-2995

Reviewer 2 Report

The paper is devoted to an exciting topic of evaluating the effect of air pollution reduction measures on carbon emissions.

The general comment is that the paper needs more detailed descriptions of the terms and methods used. More specifically:

1) Abstract: The acronyms should not be used in the abstract. The names of the models DID, PSM-DID and SDID should be changed to something more understandable. What is the (spatial) spillover effect? 
2) l. 109-110/ What are DN, DMSP/OLS, and NPP/VIIRS? Descriptions and references are missing. 
3) What is CEF_{i} in (1)? What does (44/12) mean?
4) l.148 Which website?
5) l.157 What is DID model? Some description and references are needed.
6) What is the PSM radius matching method? Some description and references are needed.
7) l. 178 How the (spatial) spillover effect is measured.  See 1) 
8) Section 3.3 Some more details should be added about the statistical procedures considered. Which model parameters are evaluated? What data is used?
9) l.207. What is a parallel trend assumption? The statistical details may be presented in the appendix.
10) l.241 What is the "Pollution heaven hypothesis"? Some description and references are needed.
11) The positive correlation of FD and UI to carbon emission should be discussed in more detail in the discussion section why it may be this way.
12) The paper is missing the list of acronyms.

Author Response

Responses to Reviewer # 2

Comments and Suggestions for Authors

The paper is devoted to an exciting topic of evaluating the effect of air pollution reduction measures on carbon emissions.

The general comment is that the paper needs more detailed descriptions of the terms and methods used. More specifically:

1) Abstract: The acronyms should not be used in the abstract. The names of the models DID, PSM-DID and SDID should be changed to something more understandable. What is the (spatial) spillover effect? 

Reply: Many thanks for your constructive suggestion, we have replaced the acronym with the full name. In this paper, the spatial spillover effect refers to the effect of APPCAP on per capita carbon emissions from its neighboring cities.

2) l. 109-110/ What are DN, DMSP/OLS, and NPP/VIIRS? Descriptions and references are missing. 

Reply: DN denotes the abbreviation of Digital Number. DMSP/OLS denotes the abbreviation of Defense Meteorological Satellite Program/Operational Linescan System. NPP/VIIRS denotes the abbreviation of National Polar-orbiting Partnership Visible Infrared Imaging Radiometer Suite.

3) What is CEF_{i} in (1)? What does (44/12) mean?

Reply: CEFi denotes the carbon content of the ith energy source, 44 denotes the relative atomic mass of carbon dioxide, 12 denotes the relative atomic mass of carbon.

4) l.148 Which website?

Reply: http://ngdc.noaa.gov/eog/dmsp/downloadV4composites.html

https://www.ngdc.noaa.gov/eog/viirs/download_dnb_composites.html

5) l.157 What is DID model? Some description and references are needed.

Reply: DID denotes the abbreviation of difference-in-differences. We have delivered the description and references in the text.

For one thing, APPCAP has caused a difference between the treatment group and the control group. For another, APPCAP has created a difference between the treatment group before and after implementing it. Thus, APPCAP can be treated as a quasi-natural experiment, and the difference-in-differences (i.e., DID) model can effectively identify the effect of it on per capita carbon emission in China [10].

6) What is the PSM radius matching method? Some description and references are needed.

Reply: The PSM radius matching method is the 1:1 logit model. The state code is as follows: psmatch2 dit x1 x2 x3 x4,out( y )logit ate neighbor(1) common caliper(.05)ties

7) l. 178 How the (spatial) spillover effect is measured.  See 1) 

Reply: The spatial spillover effect is measured by employing the spatial decomposition method, and the details are reported in the next question (see the next answer).

8) Section 3.3 Some more details should be added about the statistical procedures considered. Which model parameters are evaluated? What data is used?

Reply: The SDID approach not only identifies the average (total) effect of treatment, but also allows for decomposing the average treatment effect into both the average direct effect and the average indirect effect. The indirect effect arises from the treatment over the treated city due to neighborhood structure, as well as the effect on the untreated city closed to treated one. The general expression is shown as:

where W is the spatial weight matrix, and  denotes a specific indicator of treatment in region i at time t. Thus,  stands for the average indirect effect of the pilot region on both treated and untreated regions. Particularly, for clarity, the spatial weight matrix W can be decomposed into four parts:

where

, and

where  is an matrix with  in the main diagonal and zeros elsewhere, and , with a vector of 1’s. In this way,  stands for the neighboring effects of the city j on the city i, and i, j = T (treated) or NT (untreated). Since  and  are 0-vectors, only the other two parts are incorporated into the equation.

9) l.207. What is a parallel trend assumption? The statistical details may be presented in the appendix.

Reply: Many thanks for your constructive suggestion, the parallel trend assumption indicates that, before the implementation of the APPCAP, the per capita carbon emission of each pilot city should maintain a relatively steady trend of change. Since it’s a consensus in the policy evaluation based on the DID model, we have added this content in the text.

10) l.241 What is the "Pollution heaven hypothesis"? Some description and references are needed.

Reply: The "Pollution heaven hypothesis" holds the viewpoint that enterprises in pollution-intensive industries tend to transferred into the countries or regions with relatively low environmental standards.

11) The positive correlation of FD and UI to carbon emission should be discussed in more detail in the discussion section why it may be this way.

Reply: Many thanks for your constructive suggestion, we have added a detail discussion as follows:

On one hand, the improvement in production efficiency caused by fiscal decentrilization and industrial upgrading may lead to the reduction of per capita carbon emission. On the other hand, the rise in production scale caused by fiscal decentrilization and industrial upgrading may lead to the energy rebound effect and cause the increase of per capita carbon emission finally. Thus, the insignificant coefficients of FD and IU indicate that those two powers are even-handed.

12) The paper is missing the list of acronyms.

Reply: We have added them in the text.

Appendix

Acronyms

Full name

CO2

Per capita carbon emission

APPCAP

Air Pollution Prevention and Control Action Plan

DID

Difference-in-differences

PSM-DID

Propensity-score-matched difference-in-differences

SDID

Spatial difference-in-differences

DN

Digital Number

DMSP/OLS

Defense Meteorological Satellite Program/Operational Linescan System

NPP/VIIRS

National Polar-orbiting Partnership/Visible Infrared Imaging Radiometer Suite

IPCC

Intergovernmental Panel on Climate Change

FD

Fiscal decentralization

IU

Industrial upgrading

UR

Urbanization rate

FDI

Foreign direct investment

Zhenhua Zhang1, Jingxue Zhang2 and Yanchao Feng2*

1 Institute of Green Finance, Lanzhou University, Lanzhou 73000, PR China

2 Business School, Zhengzhou University, Zhengzhou 450001, PR China

* Correspondence: m15002182995@163.com; Tel.: +86-150-0218-2995

Reviewer 3 Report

The article, "Assessment on the Carbon Emission Reduction Effect of Air Pollution Prevention and Control Action Plan in China" is an interesting and  well written paper. 

I have the following remarks:

  1. The abstract portrays contradictory statement about per capita carbon emission. In my opinion, the statement should be supported by a soft reasoning of the phenomena.  
  2. Minor mistakes need thorough revision of the manuscript, like:
    1. Line 24. Article missing before ecological environment. Similar minor grammatic corrections are required. The manuscript may be proofread by a native English speaker.  
    2.  Line 27. Correction required: "should be converted" instead of "should be convert"
    3.  Line 49. Correction required: " a vital"
    4. Line 56. It should be propensity-score-matching
    5. Line 101. As the authors state that the success of the policy has been proved by many studies, more references need to be cited. 
    6. Equation 1. The reason of multiplying the equation by (44/12) needs explanation
    7. Line 203. yeart needs to be re-written similar the one in equation 4
    8. Line 218. Please avoid writing contractions like: It’s
  3.  The selection of control variables needs citation. Which past studies have used these variables? It will enhance the validity and justification for employing them in the underlying study.
  4. A serious issue that needs to be addressed in this study is the selection of sample. As stated in the manuscript, "47 prefecture-level and above cities in the Beijing-Tianjin-Hebei Region, the Yangtze River Delta Region, the Pearl River Delta Region, and the Fen-Wei Plain are set as the treatment group, other 238 prefecture-level and above cities are set as the control group." In this case, margin of error is high and hence, the confidence level is low. A suitable justification is required with reference.
  5. Line 145. Sources of data are not clearly described and referenced.
  6. Line 170. "After PSM matching, the anomalous samples were deleted, that is, no significant systematic difference between the treatment group and the control group, and the difference of per capita carbon emission can only be caused by APPCAP." A graphical representation of the data showing the distribution of propensity scores between the treated and control groups before and after matching maybe needed for better understanding. 
  7.  High values of R-squared in Tables 5,6, and 7 need justification

Author Response

Responses to Reviewer # 3

Comments and Suggestions for Authors

The article, "Assessment on the Carbon Emission Reduction Effect of Air Pollution Prevention and Control Action Plan in China" is an interesting and  well written paper. 

I have the following remarks:

The abstract portrays contradictory statement about per capita carbon emission. In my opinion, the statement should be supported by a soft reasoning of the phenomena.  

Reply: We have added the reason as follows:

Contradictory results are supported under different econometric models, which highlights the importance and necessary of comprehensive analysis.

Minor mistakes need thorough revision of the manuscript, like:

Line 24. Article missing before ecological environment. Similar minor grammatic corrections are required. The manuscript may be proofread by a native English speaker.  

Reply: Many thanks for your constructive advice, we have corrected the errors you mentioned and this paper have been proofread by two native English professors.

 Line 27. Correction required: "should be converted" instead of "should be convert"

Reply: We have corrected it.

 Line 49. Correction required: " a vital"

Reply: We have corrected it.

Line 56. It should be propensity-score-matching

Reply: We have corrected it.

Line 101. As the authors state that the success of the policy has been proved by many studies, more references need to be cited. 

Reply: Many thanks for your constructive advice. Indeed, the success of the policy has been proved by many studies. To avoid repetitive reference, we have listed several new published paper as follows:

  1. Huang, J., Pan, X.C., Guo, X.B., Li, G.X., 2018. Health impact of China's Air Pollution Prevention and Control Action Plan: an analysis of national air quality monitoring and mortality data. Lancet Planet Health 2(7), E313-E323.
  2. Li, J., Hou, L.P., Wang, L., Tang, L.N., 2021a. Decoupling Analysis between Economic Growth and Air Pollution in Key Regions of Air Pollution Control in China. Sustainability 13(12), 6600.
  3. Geng, G.N., Xiao, Q.Y., Zheng, Y.X., Tong, D., Zhang, Y.X., Zhang, X.Y., Zhang, Q., He, K.B., Liu, Y., 2019. Impact of China's Air Pollution Prevention and Control Action Plan on PM2.5 chemical composition over eastern China. Sci. China Earth Sci. 62(12), 1872-1884.
  4. Li, T.H., Ma, J.H., Mo, B., 2021b. Does Environmental Policy Affect Green Total Factor Productivity? Quasi-Natural Experiment Based on China's Air Pollution Control and Prevention Action Plan. Int. J. Env. Res. Pub. He. 18(15), 8216.

Equation 1. The reason of multiplying the equation by (44/12) needs explanation

Reply: 44 denotes the relative atomic mass of carbon dioxide, 12 denotes the relative atomic mass of carbon.

Line 203. year t needs to be re-written similar the one in equation 4

Reply: We have corrected it.

Line 218. Please avoid writing contractions like: It’s

Reply: We have corrected it.

The selection of control variables needs citation. Which past studies have used these variables? It will enhance the validity and justification for employing them in the underlying study.

Reply: Many thanks for your constructive advice, the validity and justification for employing them had been supported by our prior papers as follows:

  1. Feng, Y.C., Wang, X.H., Du, W.C., Wu, H.Y., Wang, J.T., 2019a. Effects of environmental regulation and FDI on urban innovation in China: A spatial Durbin econometric analysis. J. Clean. Prod. 235, 210-224.
  2. Feng, Y.C., Wang, X.H., Hu, S.L., 2021. Accountability audit of natural resource, air pollution reduction and political promotion in China: Empirical evidence from a quasi-natural experiment. J. Clean. Prod. 287, 125002.
  3. Feng, Y.C., He, F., 2020. The effect of environmental information disclosure on environmental quality: Evidence from Chinese cities. J. Clean. Prod. 276, 124067.

A serious issue that needs to be addressed in this study is the selection of sample. As stated in the manuscript, "47 prefecture-level and above cities in the Beijing-Tianjin-Hebei Region, the Yangtze River Delta Region, the Pearl River Delta Region, and the Fen-Wei Plain are set as the treatment group, other 238 prefecture-level and above cities are set as the control group." In this case, margin of error is high and hence, the confidence level is low. A suitable justification is required with reference.

Reply: Many thanks for your constructive advice, we have added

Line 145. Sources of data are not clearly described and referenced.

Reply: Many thanks for your constructive advice, we have clearly described the sources of data as follows:

For instance, the provincial data of energy use is derived from the China Energy Statistics Yearbook; the nighttime light brightness data of DMSP/OLS and NPP/VIIRS are derived from the corresponding websites  of  http://ngdc.noaa.gov/eog/dmsp/downloadV4composites.html and https://www.ngdc.noaa.gov/eog/viirs/download_dnb_composites.html, respectively; the list of APPCAP are collected from the 12th Five-Year Plan on Air Pollution Prevention and Control in Key Regions; while the socioeconomic data of control variables are derived from the China City Statistical Yearbook.

Line 170. "After PSM matching, the anomalous samples were deleted, that is, no significant systematic difference between the treatment group and the control group, and the difference of per capita carbon emission can only be caused by APPCAP." A graphical representation of the data showing the distribution of propensity scores between the treated and control groups before and after matching maybe needed for better understanding. 

Reply: Many thanks for your constructive advice, we have added the figure in the text as follows:

Figure 4. Covariate standardization bias test.

High values of R-squared in Tables 5,6, and 7 need justification.

Reply: Many thanks for your constructive advice, we have confirmed the reliability of R-squared in Tables 5, 6, and 7.

Zhenhua Zhang1, Jingxue Zhang2 and Yanchao Feng2*

1 Institute of Green Finance, Lanzhou University, Lanzhou 73000, PR China

2 Business School, Zhengzhou University, Zhengzhou 450001, PR China

* Correspondence: m15002182995@163.com; Tel.: +86-150-0218-2995

Round 2

Reviewer 1 Report

I appreciate the author(s) 's time and effort to respond to all the review comments and improve the manuscript. 

Author Response

Many thanks for your positive comments. 

God bless you!

Reviewer 3 Report

The authors have improved the manuscript but still it needs minute improvements:

  1. Line 55. "Contradictory results are supported under different econometric models, which highlights the importance and necessary of comprehensive analysis."

Correction is needed. "highlights" should be "highlight" and "necessary" should be replaced by "necessity." 

2. Round 1 comment (Line 24) and now Line 71 has not been amended but in the authors claim that it has been rectified.

3. Line 259 (Previously 101) the correction has been confirmed by the authors but not seen in the manuscript. Many studies............ and there is only one reference. Technically, a single study cannot represent many studies, so references may be repeated as the authors in their reply state that to avoid repetition, they only added a single reference. the references may be repeated in the manuscript. Better is to add more references.

4. Authors have explained the (44/12) in the response file but the same is not indicated in the manuscript. In my opinion, authors may add this information in the footnote.

5. The comment has not been addressed in the manuscript whereas authors claim to have it Selecting 47 and 238  prefecture level and above cities being the counterfactuals in the econometric techniques, has been reported amended in the manuscript but the same is not depicted in the manuscript.  

Author Response

Responses to Reviewer # 3

The authors have improved the manuscript but still it needs minute improvements:

  1. Line 55. "Contradictory results are supported under different econometric models, which highlights the importance and necessary of comprehensive analysis."

Correction is needed. "highlights" should be "highlight" and "necessary" should be replaced by "necessity." 

Reply: Many thanks for your constructive comment, we have changed the corresponding contents in the text.

  1. Round 1 comment (Line 24) and now Line 71 has not been amended but in the authors claim that it has been rectified.

Reply: Many thanks for your constructive comment, we have changed the corresponding contents in the text. That is, “the deterioration of ecological environment” has been replaced with “the ecological deterioration of air quality”.

  1. Line 259 (Previously 101) the correction has been confirmed by the authors but not seen in the manuscript. Many studies............ and there is only one reference. Technically, a single study cannot represent many studies, so references may be repeated as the authors in their reply state that to avoid repetition, they only added a single reference. the references may be repeated in the manuscript. Better is to add more references.

Reply: Many thanks for your constructive advice. Indeed, the success of the policy has been proved by many studies. To avoid repetitive reference, we have listed several new published paper as follows:

  1. Huang, J., Pan, X.C., Guo, X.B., Li, G.X., 2018. Health impact of China's Air Pollution Prevention and Control Action Plan: an analysis of national air quality monitoring and mortality data. Lancet Planet Health 2(7), E313-E323.
  2. Li, J., Hou, L.P., Wang, L., Tang, L.N., 2021a. Decoupling Analysis between Economic Growth and Air Pollution in Key Regions of Air Pollution Control in China. Sustainability 13(12), 6600.
  3. Geng, G.N., Xiao, Q.Y., Zheng, Y.X., Tong, D., Zhang, Y.X., Zhang, X.Y., Zhang, Q., He, K.B., Liu, Y., 2019. Impact of China's Air Pollution Prevention and Control Action Plan on PM2.5 chemical composition over eastern China. Sci. China Earth Sci. 62(12), 1872-1884.
  4. Li, T.H., Ma, J.H., Mo, B., 2021b. Does Environmental Policy Affect Green Total Factor Productivity? Quasi-Natural Experiment Based on China's Air Pollution Control and Prevention Action Plan. Int. J. Env. Res. Pub. He. 18(15), 8216.

As for other references, we have not added them in the text due to the spirit of scientific research and the old data.

  1. Authors have explained the (44/12) in the response file but the same is not indicated in the manuscript. In my opinion, authors may add this information in the footnote.

Reply: Many thanks for your constructive comment, we have changed the corresponding contents in the text. That is, 44 and 12 denote the relative atomic mass of carbon dioxide and carbon, respectively.

  1. The comment has not been addressed in the manuscript whereas authors claim to have it Selecting 47 and 238  prefecture level and above cities being the counterfactuals in the econometric techniques, has been reported amended in the manuscript but the same is not depicted in the manuscript.  

Reply: Many thanks for your constructive advice, we have added the official document and its website as follows:

  1. Ministry of Ecology and Environment. Report on the State of the Ecology and Environment in China 2019. Available online: http://english.mee.gov.cn/Resources/Reports/ (accessed on 23 September 2020).

Above are the detailed corrections we made according to your comments point by point. Thanks again for your meticulous review and valuable suggestions to improve our manuscript. We hope that the revised revision has addressed all the issues. We are looking forward to your positive response. If you have any queries, please don’t hesitate to contact me at the address below.

Zhenhua Zhang1, Jingxue Zhang2 and Yanchao Feng2*

1 Institute of Green Finance, Lanzhou University, Lanzhou 73000, PR China

2 Business School, Zhengzhou University, Zhengzhou 450001, PR China

* Correspondence: m15002182995@163.com; Tel.: +86-150-0218-2995